# A Predictable Obstacle Avoidance Model Based on Geometric Configuration of Redundant Manipulators for Motion Planning

**DOI:** 10.3390/s23104642

**Published:** 2023-05-10

**Authors:** Fengjia Ju, Hongzhe Jin, Binluan Wang, Jie Zhao

**Affiliations:** State Key Laboratory of Robotics and System, Harbin Institute of Technology, Harbin 150001, China; 20b908029@stu.hit.edu.cn (F.J.); 18b908042@stu.hit.edu.cn (B.W.); jzhao@hit.edu.cn (J.Z.)

**Keywords:** motion planning, collision avoidance, kinematics, redundant robots

## Abstract

When a manipulator works in dynamic environments, it may be affected by obstacles and may cause danger to people around. This requires the manipulator to be able to plan the obstacle avoidance motion in real time. Therefore, the problem solved in this paper is dynamic obstacle avoidance with the whole body of the redundant manipulator. The difficulty of this problem is how to model the manipulator to reflect the motion relationship between the manipulator and the obstacle. In order to describe accurately the occurrence conditions of the collision, we propose the triangular collision plane, a predictable obstacle avoidance model based on the geometric configuration of the manipulator. Based on this model, three cost functions, including the cost of the motion state, the cost of a head-on collision, and the cost of the approach time, are established and regarded as optimization objectives in the inverse kinematics solution of the redundant manipulator combined with the gradient projection method. The simulations and experiments on the redundant manipulator and the comparison with the distance-based obstacle avoidance point method show that our method improves the response speed of the manipulator and the safety of the system.

## 1. Introduction

In the process of task execution, such as parts assembly, object grasping, and space station construction, the manipulator is often affected by the limitation and change in the operation space and has contact and non-contact interactions with the dynamic environment. Collision detection in contact interactions is usually based on dynamic models, while in non-contact interactions, collision detection is usually based on kinematic geometric models [1]. In non-contact interactions, the manipulator should have the ability to predict whether a collision will occur and react in advance. Therefore, it is necessary to design a predictable geometric collision detection model for the manipulator as the obstacle avoidance model. At present, there are many dynamic obstacle avoidance methods based on manipulator kinematics, and the essential difference between these methods lies in the different ways of constructing the obstacle avoidance model between the manipulator and the obstacle. Depending on the type of information used in the model, we divide the current methods into distance-based methods and velocity-based methods.

The distance-based methods are based on the position of the obstacle and the manipulator, such as some sampling-based methods [2,3,4] and reinforcement learning [5]. The most typical distance-based method is the obstacle avoidance point method (OAPM) [6]. The closest point on the manipulator body to the obstacle is regarded as the obstacle avoidance point, and the velocity away from the obstacle is assigned to this point. Based on the OAPM, the authors use the human skeleton recognition method to treat a human as an obstacle and realize the obstacle avoidance of the manipulator [7]. In [8], the obstacle avoidance points are obtained from the pre-selected equidistant points on the manipulator links, which ensure that the manipulator body keeps a relatively safe distance from obstacles. A variety of collision detection methods have been proposed in computer graphics [9] and used to solve trajectory planning problems by combining them with optimization methods [10,11,12,13,14,15,16]. In [12], the authors calculate the bound of the obstacle by estimating the trajectory of the moving obstacle over a short time horizon and build the cost function of the signed Euclidean distance transform between the robot sphere and the location boundary of the obstacle to realize the trajectory planning. The sweep-out volume is proposed as the obstacle avoidance model of the manipulator in a continuous time step, and the cost based on the distance between the sweep-out volume and the obstacle is calculated in [13]. A method for strictly convex hull generation is proposed in [14] and used to calculate the continuous gradient proximity distance to generate free-collision optimized postures for humanoid robots. Gilbert–Johnson–Keerthi algorithm is used in [17] to calculate the minimum distance between a manipulator and an obstacle.

In the dynamic obstacle environment, the above distance-based modeling methods lose the velocity information of obstacles and lack the ability to predict whether a collision will occur, which reduces the safety of the system. When humans move, they tend to predict the trajectory of obstacles in their field of vision. If they think the obstacle is moving towards them, they will take action to avoid it in advance, even if the obstacle is far away from them. Therefore, the motion state of obstacles should be considered in order to achieve better dynamic obstacle avoidance. Some methods based on velocity are proposed. The velocity obstacle method is widely used in obstacle avoidance motion planning of multi-agent systems [18,19,20]. The robot is modeled as a circle or sphere, and the velocity obstacle model is established based on the position and velocity information of the robot. The velocity obstacle method is also applied to the manipulator by modeling the manipulator as a series of discrete spheres along the links [21]. Additionally, dynamic movement primitive is used in [22] to realize obstacle avoidance of the end effector by adjusting the end effector velocity direction according to the position and velocity information of the end effector and the obstacle.

However, the current obstacle avoidance methods are mainly distance-based, and the model of manipulator links is simplified into a point or sphere set. The discrete model may result in the discontinuous joint velocity when the number of sampling points in the discrete model is too small and may cause a large computation when the number of sampling points is too large. Therefore, the current problem in the field of dynamic obstacle avoidance is the lack of a velocity-based continuous obstacle avoidance model to represent the geometric configuration of the manipulator.

From the perspective of topology, the triangular relationship between two adjacent links in the three-dimensional space remains unchanged, so the geometric configuration of the redundant manipulator can be described by the motion and deformation of the triangles. Therefore, the adaptive adjustment of the manipulator in the dynamic environment is transformed into the dynamic obstacle avoidance of the triangles. Instead of being discretized into the point or sphere set, the manipulator link is considered as the side of a triangle. We propose the triangular collision plane model with three kinds of cost functions and combine them with the self-motion term of the gradient projection method [5,6,7,8,23] to calculate the inverse kinematics solution of the redundant manipulator, which solves the conflict between obstacle avoidance and task execution.

This paper is organized as follows. Section 2 describes the gradient projection method, which can make the redundant manipulator track objective and avoid obstacles at the same time. In Section 3, we introduce the triangular collision plane model we proposed and the method to realize dynamic obstacle avoidance based on this model. The corresponding simulations and experiments of our method and the contrast method are shown in Section 4. Finally, the conclusion is drawn in Section 5.

## 2. Problem Formulation

When the dimension of the joint space n exceeds the dimension of the task space m, the manipulator is redundant. There are infinite configurations of redundant manipulators for the same end effector pose, which ensures the coordination of tracking objectives and obstacle avoidance.

The relationship between the joint angular velocity vector $\dot{\theta }$ of the n degrees of freedom (DOF) redundant manipulator and the end effector velocity vector $\dot{X}$ can be described by the following equation:(1)$$\dot{X}=J\dot{\theta }$$
where $\dot{X}$ is the end effector velocity vector in m-dimensional task space, $\dot{\theta }=\left({\dot{\theta }}_{1},\ldots ,{\dot{\theta }}_{n}\right)$ is the n-dimensional joint velocity vector, and J is the m×n Jacobian matrix. 

The gradient projection method is widely used to obtain the inverse kinematics of the redundant manipulator. It decouples the motion of the redundant manipulator into the end effector motion and the self-motion. The inverse kinematic solution can be calculated by the gradient projection method [23] as follows,
(2)$$\dot{\theta }=J^{+}\dot{X}+k\left(I-J^{+}J\right)\dot{\varphi }$$
where k is a gain, $J^{+}=J^{T}{\left(JJ^{T}\right)}^{-1}$ is the pseudoinverse of J, I is an n×n identity matrix, $J^{+}\dot{X}$ is the minimum norm solution of (1), $\left(I-J^{+}J\right)$ is the projection operator, $\dot{\varphi }$ is an arbitrary vector in $\dot{\theta }$-space used to achieve some additional goals, and $\left(I-J^{+}J\right)\dot{\varphi }$ is the general solution of the homogeneous linear equations, which constitutes the null space of the redundant manipulator Jacobi matrix at the same time.

The motion of the joints generated in the null space does not change the pose of the end effector, so the motion of the redundant manipulator can be decoupled into the motion of the end effector by $J^{+}\dot{X}$ in (2) and the self-motion of the manipulator by $\left(I-J^{+}J\right)\dot{\varphi }$ in (2). In order to track the objective, the desired velocity of the end effector $\dot{X}$ is defined as
(3)$$\dot{X}={\dot{X}}_{\mathrm{obj}}+K_{p}\left(X_{\mathrm{obj}}-X\right)$$
where $K_{p}$ is a gain for the pose error $X_{\mathrm{obj}}-X$, $X_{\mathrm{obj}}$ is the objective pose of the end effector, ${\dot{X}}_{\mathrm{obj}}$ is the velocity vector of the objective, and X is the pose of the end effector.

The collision cost function E(θ, A), composed of the joint angle vector θ and the obstacle A, can be built to describe the relationship between the manipulator and the obstacle. In order to reduce the collision cost E(θ, A), $\dot{\varphi }$ in the gradient projection method can be defined as
(4)$$\dot{\varphi }=-\nabla E\left(\theta ,\;A\right)$$
where ∇E(θ, A) is the gradient of the collision cost function E(θ, A).

The iterative method we used to control the motion of the manipulator is
(5)$$\theta_{d+1}=\theta_{d}+\Delta \theta_{d}$$
where $\Delta \theta_{d}={\dot{\theta }}_{d}\cdot \Delta T$ is the joint angle increment, and ΔT is the sampling time interval.

**Remark** **1.**
*The collision cost function is the key to solving the obstacle avoidance problem because it reflects the building way of the dynamic obstacle avoidance model. The traditional methods only use the distance to build the obstacle avoidance model, while our method applies the triangle element and velocity information into the model to establish the corresponding collision cost to improve the system’s safety.*


The gradient projection method introduced in Section 2 is the framework used in our dynamic obstacle avoidance method. The focus of this paper is on the innovation of the dynamic obstacle avoidance model reflected in the collision cost function. For the sake of logical rationality and coherence, we introduce our dynamic obstacle avoidance model in Section 3 after introducing the gradient projection method in Section 2.

## 3. Obstacle Avoidance Method Design

In this section, we will introduce the proposed triangular collision plane model and 3 kinds of cost functions based on this model, obtain the obstacle avoidance optimization objective by the collision cost neural network, and design the control method for dynamic obstacle avoidance of the redundant manipulator.

### 3.1. Predictable Obstacle Avoidance Model

First of all, we simplify the redundant manipulator, which is connected by the pitch joint and yaw joints alternately, as shown in Figure 1a. The obstacle is regarded as a point, and the link of the manipulator is simplified as a line segment.

We choose two adjacent links as the two sides of a triangle and connect the starting point of the first side and the endpoint of the second side to form a triangle. The motion and deformation of the triangle are determined by joint angles, so the configuration of the manipulator can be described by the pose and shape of the triangles. Therefore, the triangles established according to the manipulator configuration can be used to replace the manipulator body for obstacle avoidance motion planning. As shown in Figure 1b, for the convenience of displaying the model of the triangles built based on the manipulator configuration, an appropriate configuration is selected so that all triangles in Figure 1a are located in the same plane.

Considering the volume and different shapes of the real obstacle, we first use a bounding box surrounding it by the oriented bounding box (OBB) method. Then, we choose 8 vertices of the box and the geometric center of the box as the obstacle points to reflect the motion of the obstacle, as shown in Figure 1c. The ith triangle $t_{i}$ matches the jth obstacle point ${\mathrm{op}}_{j}$ to form a pair, $p_{i,j}\left(t_{i},{\;\mathrm{op}}_{j}\right)$.

In Figure 2, we choose the ith (i=1) triangle, shown in Figure 1a as the grey one, to introduce our obstacle avoidance model. The points and vectors in Figure 2 are relative to the base coordinate $\Sigma_{B}$ of the manipulator. We propose 3 kinds of cost functions, including the *cost of motion state* (CMS), the *cost of head-on collision* (CHOC), and the *cost of approach time* (CAT). W is the unit vector of the relative velocity between the obstacle and the centroid $O_{i}$ of the triangle, and the CMS is built by the angle $\gamma_{i}$ between W and the unit normal vector $N_{i}$ of the triangle. The obstacle along the direction of W has an intersection point $C_{i}$ with the plane P where the triangular collision plane is located. There is a position offset $\delta_{i}$ between $C_{i}$ and $O_{i}$. The CHOC is calculated by $\delta_{i}$ and constrained by the 3 sides of the triangle. $d_{i}$ is the approach distance between the position of the obstacle $X_{\mathrm{obs}}$ and the plane P. The CAT is composed of $d_{i}$ and the relative velocity between the obstacle velocity ${\dot{X}}_{\mathrm{obs}}$ and the centroid velocity ${\dot{O}}_{i}$. Based on the above 3 costs, we establish the *triangular collision plane* as the geometric obstacle avoidance model of the manipulator. More details about the model are shown as follows.

As shown in Figure 2, we define the direction of the unit vector $L_{i}$ of the ith link as pointing from the vertex $V_{i}$ to the vertex $V_{i+1}$, and the unit vector $L_{i+1}$ is defined in the same way. $N_{i}$ is determined by $L_{i}$ and $L_{i+1}$ according to the right-hand rule. The plane P divides the space into two parts. The plane P where the triangular collision plane is located is expressed as
(6)$$A_{i}x+B_{i}y+C_{i}z+D_{i}=0$$
where $A_{i}$, $B_{i}$, and $C_{i}$ are the 3 components of $N_{i}$ in x, y, and z coordinates, respectively, and $D_{i}=-\left(A_{i}x_{i}+B_{i}y_{i}+C_{i}z_{i}\right)$. $\left(x_{i},y_{i},z_{i}\right)$ is the coordinate of $V_{i}$. We define the part that $N_{i}$ points to as the front of the space, and the other part is the back. We substitute the coordinate of the obstacle $\left(x_{\mathrm{obs}},y_{\mathrm{obs}},z_{\mathrm{obs}}\right)$ into (6). If the value is positive, it means that the obstacle is in the front part; if the value is negative, it means that the obstacle is in the back part; and if the value is zero, it means the obstacle is on the plane P.

The CMS is different in the two parts. The inner product $\eta_{i}$ is calculated by
(7)$$\eta_{i}=W\cdot \left(L_{i}\times L_{i+1}\right)$$

The motion states of the obstacle relative to the triangular collision plane in each part are defined as 3 kinds based on the value of $\eta_{i}$, including the *head-on state*, the *parallel state*, and the *escaping state*. When the obstacle is in the front part, if $\eta_{i}$ is between −1 and 0 ($\gamma_{i}$ is between 90° and 180°), the obstacle is in the head-on state, and we consider that this state can increase the likelihood of the collision. If $\eta_{i}$ is between 0 and 1 ($\gamma_{i}$ is between 0° and 90°), the obstacle is in the escaping state, and we consider the likelihood of the collision occurrence to be equal to 0. If $\eta_{i}$ is 0 ($\gamma_{i}$ is 90°), the obstacle is in the parallel state and will not collide with the triangular collision plane as well. We classify the parallel state into the escaping state in the definition of the CMS. The CMS of the front part is expressed as
(8)$${\mathrm{CMS}}_{f}=\{\begin{array}{c} \alpha \cdot \eta_{i}{}^{2}\;,\;\eta_{i}\in \left[-1,0\right)\\ 0\;,\;\eta_{i}\in \left[0,1\right] \end{array}$$
where α is a gain and is usually equal to 1.

It is the opposite when the obstacle is in the back part. If $\eta_{i}$ is between −1 and 0 ($\gamma_{i}$ is between 0° and 90°), the obstacle is in the escaping state. If $\eta_{i}$ is between 0 and 1 ($\gamma_{i}$ is between 90° and 180°), the obstacle is in the head-on state. The CMS of the back part is given as
(9)$${\mathrm{CMS}}_{b}=\{\begin{array}{c} \;0\;,\;\eta_{i}\in \left[-1,0\right]\\ \alpha \cdot \eta_{i}{}^{2}\;,\;\eta_{i}\in \left(0,1\right] \end{array}$$

When the value of the CMS is not 0, it does not indicate that the collision must occur. Next, we introduce the CHOC into our model to restrict the likelihood of the collision. 

The line which passes the position of the obstacle along the direction of W is expressed as
(10)$$\left(x-x_{\mathrm{obs}}\right)/m=\left(y-y_{\mathrm{obs}}\right)/n=\left(z-z_{\mathrm{obs}}\right)/q$$
where $\left(x_{\mathrm{obs}},y_{\mathrm{obs}},z_{\mathrm{obs}}\right)$ is the coordinate of the obstacle, and m,n,q are the 3 components of W. Combing (6) and (10), we can obtain $C_{i}$ and determine whether it is in the triangle.

Considering the geometric volume of the real manipulator and the obstacle, we define the slack factor ϵ (≥1), which enlarges the triangle appropriately and proportionally. The slack vertices $V_{i}^{\prime }$, $V_{i+1}^{\prime }$, and $V_{i+2}^{\prime }$ are calculated by
(11)$$\vec{O_{i}V_{r}^{\prime }}=\epsilon \cdot \vec{O_{i}V_{r}}$$
where r is equal to i, i+1, i+2. We use the slack triangle to replace the initial triangle in the following. ϵ can directly affect the response time of the obstacle avoidance by controlling the size of the triangle, which determines the influence range of our model. The sides of the slack triangle are shown by the green dashed line in Figure 2, and the dark gray part is the slack triangular collision plane. 

For dynamic obstacle avoidance, the manipulator is replaced by triangles to plan obstacle avoidance motion. Considering the geometric volume of the manipulator links, we use a slack factor ϵ to enlarge the size of the triangle in order to make the triangles cover the manipulator links. Therefore, the manipulator link is located between the center and boundary of the triangle actually. The goal is to ensure that both the center and boundary of the triangle do not collide with obstacles. Therefore, when constructing the CHOC, set the value of CHOC at the center of the triangle to the maximum. It makes the obstacle have a motion direction from the center of the triangle to the outside of the boundaries relative to the triangle.

In each control cycle, $O_{i}$ has a distance to each vertex of the slack triangle, and among them, the largest one is defined as the maximum distance, $\mu_{i}$. The closer $O_{i}$ and $C_{i}$ are, the higher the possibility of collision occurrence. The CHOC is defined as
(12)$$\mathrm{CHOC}=\{\begin{array}{c} \beta \cdot {\left(\delta_{i}/\mu_{i}-1\right)}^{2}\cdot e^{-\lambda \cdot \Delta t},\;\mathrm{if}\;C_{i0}\;\mathrm{is}\;\mathrm{in}\;\mathrm{the}\;\mathrm{triangle}\\ \beta \cdot {\left(\delta_{i}/\mu_{i}-1\right)}^{2}\cdot \left(1-e^{-\lambda \cdot \Delta t}\right),\;\mathrm{otherwise} \end{array}$$
where β is a gain, Δt is the duration beginning from the obstacle entering or leaving the triangle boundary and is equal to 0 initially, λ is a smooth factor, and $C_{i0}$ is $C_{i}$ in the initial state of the obstacle. The calculation of CHOC is chosen by the relative position between $C_{i0}$ and the triangle. If $C_{i}$ is in the triangle, it is possible for the obstacle to collide with the triangular collision plane, and CHOC is not equal to 0; otherwise, if $C_{i}$ is outside the triangle or $C_{i}$ does not exist when the obstacle is in a parallel state relative to the triangular collision plane, the collision will not happen, and CHOC is equal to 0. The circular contour makes the calculation simple in (12), but the shape difference between the circular cost contour and triangular boundary causes a great change in the CHOC value at the triangular boundary on the space scale. As for this, $e^{-\lambda \cdot \Delta t}$ is added to smooth the cost at the triangular boundary on the time scale and only changes when $C_{i}$ crosses the boundary. 

We consider that when the obstacle is in the head-on state, and $C_{i}$ is in the slack triangle, the obstacle will collide with the triangular collision plane as long as the obstacle keeps the same velocity direction. Then, we use the CAT to describe the time urgency of the collision when the obstacle is approaching the triangular collision plane. It is defined as
(13)$$\mathrm{CAT}=\rho ∥{\dot{X}}_{\mathrm{obs}}-{\dot{O}}_{i}∥/d_{i}$$
where ρ is a gain. When the distance is large, the CAT is not 0. The distance-based obstacle avoidance methods usually require setting a safe distance and do not avoid obstacles beyond a certain safe distance. However, for dynamic obstacle avoidance, the safe distance is often difficult to determine, and factors such as the velocity of the obstacle need to be considered. We do not need to set a safe distance in our obstacle avoidance model by using the distance between the obstacle and the triangular collision plane to calculate CAT. The farther the distance, the lower the value of CAT, which ensures the obstacle avoidance model works without limitation by the safe distance.

The above 3 cost functions represent the occurrence conditions of the collision. Only if all 3 conditions are satisfied will the collision occur; that is, CMS, CHOC, and CAT are not equal to 0 at the same time. Therefore, the collision cost function $E_{i,j}$ between the ith triangular collision plane and the jth obstacle point should be composed of the multiplication of the above 3 cost functions and is defined as
(14)$$E_{i,j}\left(\theta ,{\;\mathrm{op}}_{j}\right)=\mathrm{CMS}\cdot \mathrm{CHOC}\cdot \mathrm{CAT}$$

The calculation process of $E_{i,j}$, shown in the flow chart in Figure 3a, can be considered as a collision cost neuron $\Gamma \left(p_{i,j}\right)$ with the input $p_{i,j}\left(t_{i},\;{\mathrm{op}}_{j}\right)$ and the output $E_{i,j}$.

The collision cost neural network to calculate the optimization objective E is constructed, as shown in Figure 3b. The inputs of the network are triangles and obstacle points, and the output is the total cost. The maximal $E_{i,j}$ between one triangle plane collision and all obstacle points is regarded as $E_{i,\max}$. We add all $E_{i,\max}$ by weights $w_{i}$ to obtain the total cost E,
(15)$$E=\sum_{i=1}^{s}w_{i}E_{i,\max}$$
where s is the number of the triangular collision plane. The weights $w_{i}$ are set to 1 in our model, which means that the weight of the impact of each triangular collision plane on the obstacle avoidance motion is the same.

The control diagram of the manipulator system with tracking and obstacle avoidance is also shown in Figure 3b. The input of the system is the obstacle point position $X_{\mathrm{obs}}$ and the objective pose $X_{\mathrm{obj}}$ of the end effector, and the output $U_{\mathrm{out}}$ is the command of the joint angles.

For our model, the setting of triangles for the configuration of the manipulator is also important. The more triangles divided, the more accurate the obstacle avoidance model. However, an increase in the number of triangles can lead to a decrease in computational efficiency. Therefore, there is a trade-off between the model accuracy and the computational efficiency. In the process of modeling, we consider the length of manipulator links, and if the link is short enough, it can be ignored in order to improve computational efficiency. However, the ignored link can be covered by the adjustment of the slack factor ϵ in (11).

### 3.2. Modeling Strategy for Singularity

When the manipulator and the obstacle are in one of the following two situations, we consider that the obstacle avoidance model reaches the singular state:
(1)The obstacle is nearly moving on plane P in Figure 2; that is, $d_{i}$ is close to 0, and the obstacle is in parallel state. (2)The two adjacent links are close to collinear; that is, $\theta_{i}$ is close to 180°. In this case, the triangular collision plane cannot be formed.


In order to make our model work in the singular state, we simplify the triangular collision plane based on two adjacent links into the *collision line segment* based on each link shown as the green dashed line in Figure 4. The gray plane $P^{\prime }$ is composed of the ith link and the obstacle position, and all variables in the model are represented on this plane. $N_{i}$ is the unit normal vector of the link on the plane now. W is determined by the projection of the obstacle velocity and centroid velocity on the plane $P^{\prime }$. The slack vertices $V_{i}^{\prime }$ and $V_{i+1}^{\prime }$ replace the initial vertices, and $\mu_{i}$ is half of the slack link length. $C_{i}$ is the intersection point between the line which passes the obstacle along the direction of W and the line where the collision line segment is located. $d_{i}$ is the approach distance between the position of the obstacle and the collision line segment. The CMS and CAT of the collision line segment are the same as that of the triangular collision plane. There is no shape difference, so $e^{-\lambda \cdot \Delta t}$ can be removed in CHOC. The obstacle avoidance motion is performed on the plane $P^{\prime }$, and this model is used only when the triangular collision plane model is close to the singular state.

## 4. Simulation and Experiment

We perform simulations and experiments on ABB Yumi with two arms with seven DOF. In order to verify our model and compare it with distance-based dynamic obstacle avoidance methods, we choose the obstacle avoidance point method (OAPM) [7], which is representative of the contrast method. For the sake of fairness, we design the same obstacle avoidance experiment as the one in [7]. Two sets of simulations and experiments are performed to compare the triangular collision plane method (TCPM) with the OAPM. Because of the limitation of the degrees of freedom of the ABB Yumi, it cannot maintain both the position and the orientation of the end-effector and avoid obstacles at the same time. In order to better show the obstacle avoidance effect, in the one-triangle case, we make the end-effector maintain the position, while in the two-triangle case, we make the end-effector maintain the orientation. The objective velocity ${\dot{X}}_{\mathrm{obj}}$ in the one-triangle/two-triangle case is set to the three-dimensional zero vector. A moving objective does not have an effect on obstacle avoidance. Based on the redundancy and the gradient projection method from Equation (2), the motion of obstacle avoidance generated in the null space does not influence the motion of task execution for tracking a moving objective. The purpose of setting the objective velocity to zero is just convenient to show the obstacle avoidance effect.

In order to compare clearly, we use a yellow ping-pong ball as the dynamic obstacle. It can be considered as only one obstacle point. In the simulation, the obstacle with accelerated rectilinear motion approaches the manipulator. In the experiment, the vision system uses ZED 2i camera to identify the obstacle in the environment and uses the Kalman filter to process the obstacle position information.

In the OAPM, the obstacle avoidance point is defined as the closest point on the manipulator body to the obstacle. The distance between them is called critical distance, $d_{o}$. Using the unit vector $n_{o}$, which points from the obstacle to the obstacle avoidance point, and combining with the gradient projection method to solve inverse kinematics, the exact solution with reduced operational space is proposed in [7] as
(16)$$\dot{\theta }=J^{+}\dot{X}+\alpha_{\eta }{\left[J_{d_{o}}\left(I-J^{+}J\right)\right]}^{+}\left({\dot{x}}_{o}-J_{d_{o}}J^{+}\dot{X}\right)$$
where $J^{+}$ is the pseudoinverse of the J associated with the end effector, $\dot{X}$ is the m×1 desired velocity of the end effector calculated from (3), $J_{d_{o}}=n_{o}{}^{T}J_{o}$ is a 1×n matrix, $J_{o}$ is the Jacobian matrix associated with the obstacle avoidance point, ${\dot{x}}_{o}=\alpha_{o}\cdot v_{o}$ is the desired obstacle avoidance point velocity, $v_{o}$ is the nominal velocity, and $\alpha_{o}\left(d_{o}\right)$ and $\alpha_{\eta }\left(d_{o}\right)$ are the gains related to the critical distance and limited by the distance of influence range $d_{i}$ and the unity gain distance $d_{m}$.

### 4.1. One-Triangle Case

The right arm of ABB Yumi without a hand is used in the one-triangle case. We select the shoulder, elbow, and wrist, namely the coordinate origins of the first, third, and fifth joints, as the three vertices of the triangle. The lines connecting the selected vertices are considered as the triangle sides to form a triangle. In this simulation and experiment, the manipulator avoids obstacles and maintains the position of the end effector at the same time. The screenshots per second of the simulations and experiments are shown in Figure 5a,b.

Comparing the starting time when the joint angles, velocities, and accelerations change by the two methods in Figure 5c,e, we can see that the time of obstacle avoidance motion generation by the TCPM is earlier than that by the OAPM. It proves that the TCPM can predict, take action in advance and improve the safety of the system. The changes in the joint angles, velocities, and accelerations in experiments shown in Figure 5d,f are the same as the simulations. From the comparison, it can be seen that the accelerations of the TCPM are smoother than the ones of the OAPM and with fewer oscillations. When the obstacle is captured by the vision system, the triangular collision plane model is used to determine whether the obstacle will collide in the current motion state. If a collision occurs, obstacle avoidance action will be taken; otherwise, there is no action. In the beginning, due to the long distance and low obstacle velocity, the effect of obstacle avoidance is weak. However, as the obstacle approaches, the effect is gradually enhanced. It can be seen from Figure 5g–i that when the intersection point $C_{i}$ is out of the slack triangle, the CHOC and total cost smoothly drop to 0 by the effect of $e^{-\lambda \cdot \Delta t}$, and the manipulator completes the obstacle avoidance motion. Then, as long as the obstacle maintains the direction of velocity, collision will not happen, even if the approach distance is shortening and the CAT increases with the motion of the obstacle. Figure 5i shows the changes in the initial and slack triangular collision plane and the intersection points.

It is necessary to expand $d_{i}$ and $d_{m}$ to adapt to the dynamic obstacle in OAPM. However, the expansion of the distance range reduces the standard of collision. It can lead to failure in accurately determining whether a collision will occur and affect the task execution of the manipulator. We equidistantly select 105 discrete points on the manipulator links to calculate the minimum distance between them and the obstacle. The number of discrete points should be set in an appropriate range to trade off the computation and the smoothness of obstacle avoidance velocity. Figure 5j shows the change in the critical distance between the two methods. Due to the prediction made in advance for obstacle avoidance, the TCPM makes the manipulator keep a larger critical distance in the process of obstacle approaching than the OAPM.

### 4.2. Two-Triangle Case

We apply our method to the two triangular collision planes case. We use the left arm of ABB Yumi with a two-finger hand and add the center of the hand as the fourth vertex. The shoulder, elbow, wrist, and hand are connected in turn to form two triangles. The first triangular collision plane is composed of the shoulder, elbow, and wrist, shown in blue lines in Figure 6a, and the second one is composed of the elbow, wrist, and hand, shown in magenta lines. 

Considering the limit due to the number of DOF and the show effect, we make the manipulator only maintain the orientation of the end effector in this simulation and experiment. It can be seen from Figure 6 that the arm grasping a cup can avoid the obstacle and maintain the orientation of the end effector at the same time. At the beginning of the simulation and experiment, due to the approach of the obstacle, the CATs in the two triangular collision planes increase. It leads to an increase in total costs for both planes. However, with the gradually enhancing effect of our method, the total costs decrease in the end due to the CMSes and CHOCs, shown in Figure 6g–j. From the comparison between Figure 6c,e, it can be seen that our method can make earlier obstacle avoidance motion and hold a larger critical distance in the end, as shown in Figure 6k. The results show that our method also works in this case, and the obstacle avoidance model can be extended into the multiple triangles case with the increase in the links.

The average computation time of the TCPM is approximately 50 ms (one-triangle case) and 60 ms (two-triangle case) per iteration on a single CPU. The average computation time of the OAPM is approximately 1 ms on the same computing equipment. The computation time of most of the obstacle avoidance planning methods approximately lies between 1 ms and 2 s [13,15]. There is a trade-off between computational efficiency and dynamic obstacle avoidance performance. We sacrifice a certain level of computational efficiency to achieve predictable obstacle avoidance by considering the velocity of obstacles. The above simulations and experiments can verify our method works well in dynamic obstacle avoidance.

The value of the parameters in the simulation and experiment are presented in Table 1.

## 5. Conclusions

In this paper, we bring the triangle element and velocity information into the dynamic obstacle avoidance model based on the geometric configuration of the redundant manipulator. We propose a predictive obstacle avoidance model, the triangular collision plane, build three kinds of cost functions and apply this model to the gradient projection method for the coordination between obstacle avoidance and task execution. The collision cost neural network is constructed by the collision cost neuron to obtain the collision cost. When the CMS, CHOC, and CAT are not equal to 0 at the same time, the collision will occur, and obstacle avoidance motion is generated. This model has the ability to accurately predict the collision based on the motion of the obstacle, realizes the active obstacle avoidance of the manipulator, and improves the safety of the system. The other advantage is that the geometric continuity of our model ensures the smoothness of the obstacle avoidance motion without considering the sampling number of the discrete model. Both the simulations and the experiments in this paper show the characteristics of our methods. In future work, we will try to apply our method to multi-task coordination to further improve the system safety, and we will also try to shorten computation time from the aspect of further optimizing models, improving the code, and using the computing equipment with higher computation performance.

## Figures and Tables

**Figure 1 sensors-23-04642-f001:**
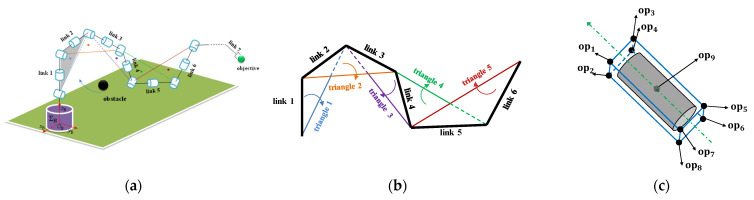
The simplified manipulator and obstacle in the triangular collision plane method. (**a**) The simplified manipulator and obstacle models; (**b**) the triangles in the same plane for demonstration; (**c**) the obstacle model.

**Figure 2 sensors-23-04642-f002:**
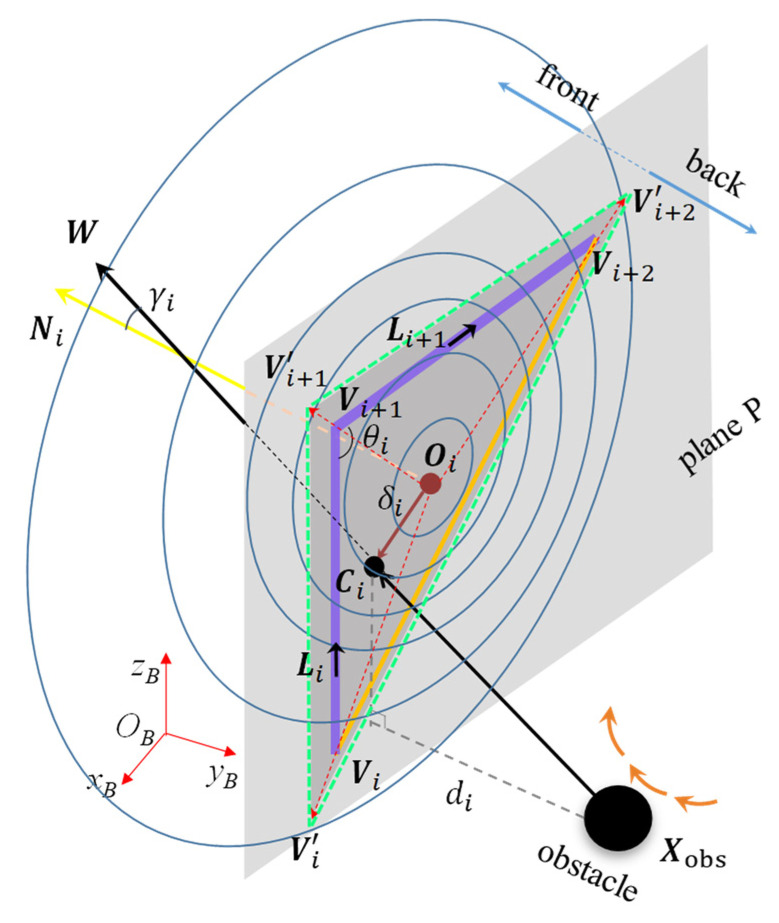
The triangular collision plane model.

**Figure 3 sensors-23-04642-f003:**
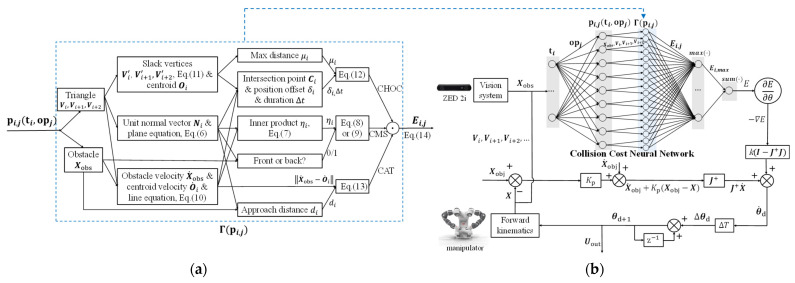
The triangular collision plane method for dynamic obstacle avoidance. (**a**) The collision cost neuron; (**b**) the control diagram of the manipulator system.

**Figure 4 sensors-23-04642-f004:**
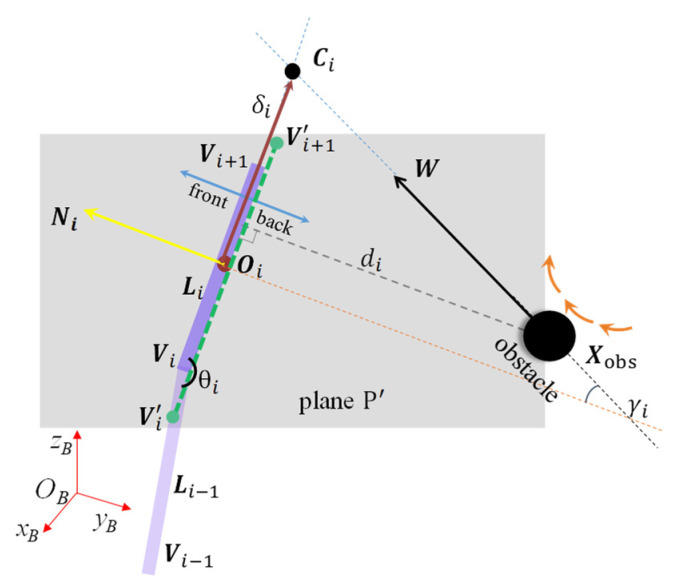
The collision line segment model.

**Figure 5 sensors-23-04642-f005:**
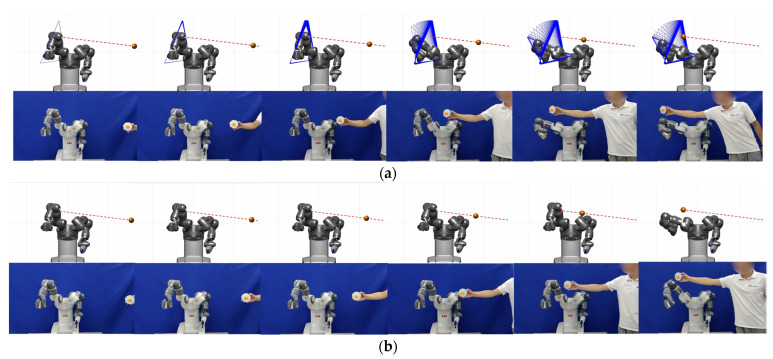
The comparison of TCPM and OAPM on ABB Yumi in the one-triangle case. (**a**) The simulation and experiment screenshots per second by TCPM; (**b**) the simulation and experiment screenshots per second by OAPM; (**c**) the joint angles, velocities, and accelerations in the simulation by TCPM; (**d**) the joint angles, velocities, and accelerations in the experiment by TCPM; (**e**) the joint angles, velocities, and accelerations in the simulation by OAPM; (**f**) the joint angles, velocities, and accelerations in the experiment by OAPM; (**g**) sub and total costs in the simulation by TCPM; (**h**) sub and total costs in the experiment by TCPM; (**i**) the change process of the triangular collision plane; (**j**) the comparison of the critical distance between TCPM and OAPM.

**Figure 6 sensors-23-04642-f006:**
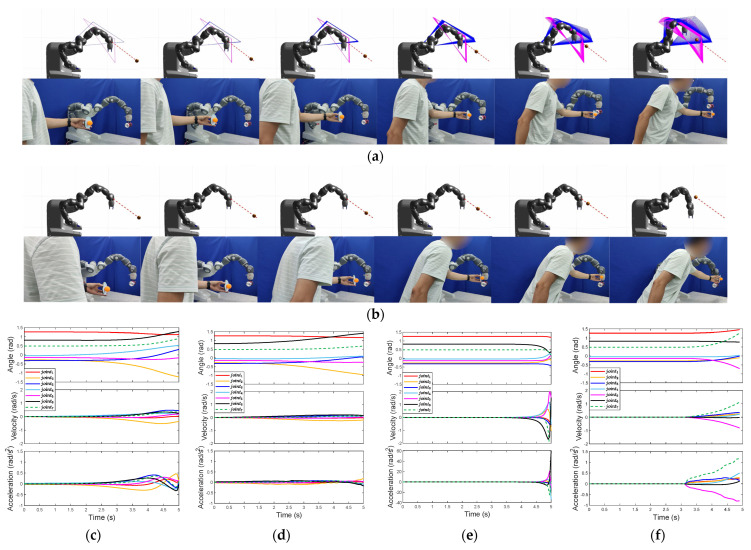
The comparison of TCPM and OAPM on ABB Yumi in the two-triangle case. (**a**) The simulation and experiment screenshots per second by TCPM; (**b**) the simulation and experiment screenshots per second by OAPM; (**c**) the joint angles, velocities, and accelerations in the simulation by TCPM; (**d**) the joint angles, velocities, and accelerations in the experiment by TCPM; (**e**) the joint angles, velocities, and accelerations in the simulation by OAPM; (**f**) the joint angles, velocities, and accelerations in the experiment by OAPM; (**g**) sub costs in the simulation by TCPM; (**h**) total costs in the simulation by TCPM; (**i**) sub costs in the experiment by TCPM; (**j**) total costs in the experiment by TCPM; (**k**) the comparison of the critical distance between TCPM and OAPM.

**Table 1 sensors-23-04642-t001:** Parameters used in the simulations and experiments.

Parameters	*k*	α	β	ρ	ϵ	$v_{o}$	$K_{p}$	$d_{i}$	$d_{m}$	$\Delta T_{sim}$	$\Delta T_{exp}$	λ
Value	10	1	1	1	2.0	0.03 m/s	1.0	0.8 m	0.6 m	0.05 s	0.1 s	10

## Data Availability

The data is unavailable due to privacy.

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
