# Peer review of "A Predictable Obstacle Avoidance Model Based on Geometric Configuration of Redundant Manipulators for Motion Planning"

_sensors, 2023, doi:10.3390/s23104642_

Round 1

Reviewer 1 Report

This paper is about the obstacle avoidance of redundant manipulators.

Neither the abstract nor the conclusions lay out the article's main findings. Literature references do not reflect the accurate review related to a topic. Rather than associated latest findings of obstacle avoidance by manipulators.

The approach in section two is weak.

The main problem is that authors introduce their own new notations, while true scientific research starts from the investigation of what is a state of the art in the patents, books, and scientific publications. And not necessarily in the terms, the authors speak.

I do not think this paper is valid for publication at the moment.

Reviewer 2 Report

This paper presents an interesting study of using a novel predictable obstacle avoidance model for redundant manipulator motion planning. The paper is well-written and the technical part is sound. The reviewer has the following comments:

Major:

(1) The number of triangles used significantly influences the collision model accuracy and the computing efficiency. Therefore, the study of (i) triangles vs accuracy and (ii) triangles vs efficiency should be included to further demonstrate the potential of the proposed method.

Minor:

(1) Check the affiliation information

(2) The quality of Figure5(i) should be improved 

Reviewer 3 Report

The paper is very well written, and contributes a predictable obstacle avoidance model for motion planning, which enables to improve of the response speed of the manipulator and the safety of the system. I recommend that this manuscript be accepted after a minor revision.

Reviewer 4 Report

1. It is mentioned that the tracking velocity of the end-effector is related to the velocity of the objective from equation (3). Seen from the provided demo video, the position of the objective seems to be fixed in the first 2 examples, while in the 3rd example the position of the end-effector changes. Please specify how the objective velocity is defined in the demo video, and does a moving objective have an effect on obstacle avoidance?

2. It is concluded that the TCPM can predict, take action in advance and improve the safety of the system. I think when to start taking action on obstacle avoidance is also related to the parameters chosen, such as ?, ?, ?, etc. Please analyze if the change of these parameters will affect the response time of obstacle avoidance

3. For another conclusion that the geometric continuity of our model ensures the smoothness of the obstacle avoidance motion without considering the sampling number of the discrete model, please provide additional materials to explain the smothness.

4. It is well known that for real-time obstacle avoidance of robots, computational efficiency is an issue that cannot be ignored. Will the computational efficiency of the proposed TCPM be better than other methods. Please provide detailed data or explanation.

Reviewer 5 Report

In this paper, the authors present an obstacle avoidance method for redundant manipulators. The paper is well structured and results seems solid, but there are some issues that should be addressed to make it more understandable and clear.

Figure 1 under c) has text "The obstacle points of an obstacle.", which should be changed

In section 3 there is a sentence "The obstacle avoidance motion planning of the manipulator body is transformed into the obstacle avoidance motion planning of the whole triangles." It is not clear what are whole triangles.

Sentence "As shown in Fig. 1(b), an appropriate configuration is selected so that all triangles in Fig. 1(a) are located in a same plane." Does this mean that the robot must always be configured so that all triangles are in the same plane? That would severely limit the possible movement of the robot. This should be explained in more detail

Line 171 "The points and vectors in Fig. 2 are relative to the base coordinate ΣB of the manipulator." Where is ΣB in Figure 1?

Line 195 "When the obstacle is in the front part,..." There should be mathematical explanation when obstacle is in the front part.

Line 228 "The closer between ??  and ??, the higher possibility of the collision occurrence is". This should be explained further, because the sides of the triangle are the links of the robot and not the center of the triangle.

Line 248 "It ensures the model is not limited by the range of distance." This sentence is not clear and should be further explained.

In equation (15), how are weights wi determined?

Line 289 "The obstacle avoidance motion is performed on the plane P' and this model is only used in singular state". Do you use this only for singularity or close to the singularity, 

Line 365 "Simulation B shows our method applied to the two triangular collision planes case.", What is simulation B and is there simulation A? 

The comparison between static and dynamic methods of obstacle avoidance seems to be an unfair comparison. The method should be compared with the other methods of dynamic obstacle avoidance.

Round 2

Reviewer 4 Report

I think the revised manuscript to be a well-organised paper that meets the requirements for publication in this journal. Thus, I no longer have any further suggestion.

Reviewer 5 Report

The authors addressed all the problems and improved the quality of the paper.